# Is GPT-4 a Good Data Analyst?

**Liying Cheng**[1,2]    **Xingxuan Li** [*1,3]    **Lidong Bing**[1,2]
[1]DAMO Academy, Alibaba Group, Singapore   [2]Hupan Lab, 310023, Hangzhou, China
[3]Nanyang Technological University, Singapore
{liying.cheng, xingxuan.li, l.bing}@alibaba-inc.com

## Abstract

As large language models (LLMs) have demonstrated their powerful capabilities in plenty of domains and tasks, including context understanding, code generation, language generation, data storytelling, etc., many data analysts may raise concerns if their jobs will be replaced by artificial intelligence (AI). This controversial topic has drawn great attention in public. However, we are still at a stage of divergent opinions without any definitive conclusion. Motivated by this, we raise the research question of "*is GPT-4 a good data analyst?*" in this work and aim to answer it by conducting head-to-head comparative studies. In detail, we regard GPT-4 as a data analyst to perform end-to-end data analysis with databases from a wide range of domains. We propose a framework to tackle the problems by carefully designing the prompts for GPT-4 to conduct experiments. We also design several task-specific evaluation metrics to systematically compare the performances between several professional human data analysts and GPT-4. Experimental results show that GPT-4 can achieve comparable performance to humans. We also provide in-depth discussions about our results to shed light on further studies before reaching the conclusion that GPT-4 can replace data analysts. Our code, data and demo are available at: https://github.com/DAMO-NLP-SG/GPT4-as-DataAnalyst.

## 1 Introduction

Large language models (LLMs) such as OpenAI's GPT series have shown their strong abilities on various tasks in the natural language processing (NLP) community, including data annotator (Ding et al., 2023), data evaluator (Chiang and Lee, 2023; Luo et al., 2023; Wang et al., 2023; Wu et al., 2023b), etc. Beyond NLP tasks, researchers also evaluate the LLM abilities in multiple domains, such as

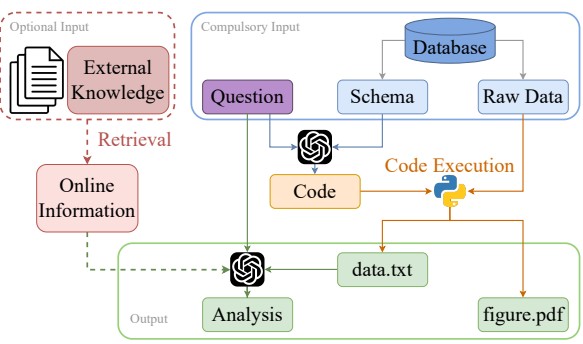

Figure 1: A figure showing the flow of our proposed framework regarding GPT-4 as a data analyst. The compulsory input information containing both the business question and the database is illustrated in the blue box on the upper right. The optional input referring to the external knowledge source is circled in the red dotted box on the upper left. The outputs including the extracted data (i.e., *"data.txt"*), the data visualization (i.e., *"figure.pdf"*) and the analysis are circled in the green box at the bottom.

finance (Wu et al., 2023c), healthcare (Han et al., 2023; Li et al., 2023b), biology (Zheng et al., 2023), law (Sun, 2023), psychology (Li et al., 2023a), etc. Most of these researches demonstrate the effectiveness of LLMs when applying them to different tasks. However, the strong ability in understanding, reasoning, and creativity causes some potential anxiety among certain groups of people.

As LLMs are introduced and becoming popular not only in the NLP community but also in many other areas, those people in and outside of the NLP community are considering or worrying whether artificial intelligence (AI) can replace certain jobs (Noever and Ciolino, 2023; Wu et al., 2023a). One such job role that could be naturally and controversially "replaced" by AI is data analyst (Tsai et al., 2015; Ribeiro et al., 2015). The main and typical job scopes for a data analyst include extracting relevant data from several databases based on business partners' requirements, presenting data visualization in an easily understandable way, and also pro-

---

*Xingxuan Li is under the Joint Ph.D. Program between Alibaba and Nanyang Technological University.

viding data analysis and insights for the audience. This job involves a relatively routine scope, which may become repetitive at times. It also requires several technical skills, including but not limited to SQL, Python, data visualization, and data analysis, making it relatively expensive. As this job scope may adhere to a relatively fixed pipeline, there is a heated public debate about the possibility of an AI tool replacing a data analyst, which attracts significant attention.

In this paper, we aim to answer the following research question: *Is GPT-4 a good data analyst?* To answer this question, we conduct preliminary studies on GPT-4 to demonstrate its potential capabilities as a data analyst. We quantitatively evaluate the pros and cons of LLM as a data analyst mainly from the following metrics: performance, time, and cost. Specifically, we treat GPT-4 (gpt-4-0314)[1] as a data analyst to conduct several end-to-end data analysis problems. The flow of our proposed framework is shown in Figure 1. According to the given question, the model has to identify the relevant tables and schemes in the databases that contain the necessary data, and then extract the data from the databases and organize it in a way that is suitable for figure generation. Then, it is required to analyze the data to identify trends, patterns, and insights that can help answer the initial question. Since there is no existing dataset for such data analysis problems, we choose one of the most related datasets NvBench (Luo et al., 2021) , and add the data analysis part on top. We design several automatic and human evaluation metrics to comprehensively evaluate the quality of the data extracted, charts plotted and data analysis generated.

Experimental results show that GPT-4 can beat an entry-level data analyst and an intern data analyst in terms of performance and have comparable performance to a senior-level data analyst. In terms of the cost and time of our experiments, GPT-4 is much cheaper and faster than hiring a data analyst. However, since it is a preliminary study on whether GPT-4 is a good data analyst, we conduct some additional experiments and provide fruitful discussions on whether the conclusions from our experiments are reliable in real-life business from several perspectives, such as whether the questions are practical, whether the human data analysts we choose are representative, etc. These results sug-

gest further studies are needed before concluding whether GPT-4 is a good data analyst. To summarize, our contributions include:

- We for the first time raise the research question of whether GPT-4 is a good data analyst, and quantitatively evaluate the pros and cons. However, further research is still required to reach a definitive conclusion.

- For such a typical data analyst job scope, we propose an end-to-end automatic framework to conduct data collection, visualization, and analysis.

- We conduct a systematic and professional human evaluation of GPT-4's outputs. The data analysis and insights with good quality can be considered as the first benchmark for data analysis in the NLP community.

## 2 Related Work

### 2.1 Related Tasks and Datasets

Since our task setting is new in the NLP community, there is no existing dataset that is entirely suitable for our task. We explore the most relevant tasks and datasets. First, the NvBench dataset (Luo et al., 2021) translates natural language (NL) queries to corresponding visualizations (VIS), which covers the first half of the main job scope of a data analyst. This dataset has 153 databases along with 780 tables in total and covers 105 domains, and this task (NL2VIS) has attracted significant attention from both commercial visualization vendors and academic researchers. Another popular subtask of the NL2VIS task is called text-to-SQL, which converts natural language questions into SQL queries (Zhong et al., 2017; Guo et al., 2019; Qi et al., 2022; Gao et al., 2022). Spider (Yu et al., 2018), SParC (Yu et al., 2019b) and CoSQL (Yu et al., 2019a) are three main benchmark datasets for text-to-SQL tasks. Since this work is more focused on imitating the overall process of the job scope of a data analyst, we adopt the NL2VIS task which has one more step forward than the text-to-SQL task.

For the second part of data analysis, we also explore relevant tasks and datasets. Automatic chart summarization (Mittal et al., 1998; Ferres et al., 2013) is a task that aims to explain a chart and summarize the key takeaways in the form of natural language. Indeed, generating natural language summaries from charts can be very helpful to infer key insights that would otherwise require a lot

---

[1]The most advanced model in the GPT series at the time of this paper was written.

of cognitive and perceptual effort. In terms of the dataset, the chart-to-text dataset (Kantharaj et al., 2022) aims to generate a short description of the given chart. This dataset also covers a wide range of topics and chart types. Another relevant NLP task is called data-to-text generation (Gardent et al., 2017; Dušek et al., 2020; Koncel-Kedziorski et al., 2019; Cheng et al., 2020). However, the output of all these existing works is descriptions or summaries in the form of one or a few sentences or a short paragraph. In contrast, data analysts are required to provide more insightful comments instead of intuitive summaries. Furthermore, in the practical setting of data analytics work, one should highlight the analysis and insights in bullet points to make them clearer to the audience. Therefore, in this work, we aim to generate the data analysis in the form of bullet points instead of a short paragraph.

## 2.2 Abilities of GPT-3, ChatGPT and GPT-4

Researchers have demonstrated the effectiveness of GPT-3 and ChatGPT on various tasks (Ding et al., 2023; Chiang and Lee, 2023; Shen et al., 2023; Luo et al., 2023; Wang et al., 2023; Wu et al., 2023b; Li et al., 2023a; Han et al., 2023; Li et al., 2023b). For example, Ding et al. (2023) evaluated the performance of GPT-3 as a data annotator. Their findings show that GPT-3 performs better on simpler tasks such as text classification than more complex tasks such as named entity recognition (NER). Wang et al. (2023) treated ChatGPT as an evaluator. They used ChatGPT to evaluate the performance of natural language generation (NLG) and to study its correlations with human evaluation. They found that the ChatGPT evaluator has a high correlation with humans in most cases, especially for creative NLG tasks.

GPT-4 is proven to be a significant upgrade over the existing models, as it is able to achieve more advanced natural language processing capabilities (OpenAI, 2023). For instance, GPT-4 is capable of generating more diverse, coherent, and natural language outputs. It is also speculated that GPT-4 may be more capable of providing answers to complex and detailed questions and performing tasks requiring deeper reasoning and inference (Bubeck et al., 2023). These advantages will have practical implications in various industries, such as customer service, finance, healthcare, and education, where AI-powered language processing can enhance communication and problem-solving. In this work, we regard GPT-4 as a data analyst to conduct our experiments.

## 3 GPT-4 as a Data Analyst

### 3.1 Background: Data Analyst Job Scope

The main job scope of a data analyst involves utilizing business data to identify meaningful patterns and trends from the data and provide stakeholders with valuable insights for making strategic decisions. To achieve their goal, they must possess a variety of skills, including SQL query writing, data cleaning and transformation, visualization generation, and data analysis.

To this end, the major job scope of a data analyst can be split into three steps based on the three main skills mentioned above: data collection, data visualization and data analysis. The initial step involves comprehending business requirements and deciding which data sources are pertinent to answering them. Once the relevant data tables have been identified, the analyst can extract the required data via SQL queries or other extraction tools. The second step is to create visual aids, such as graphs and charts, that effectively convey insights. Finally, in the data analysis stage, the analyst may need to ascertain correlations between different data points, identify anomalies and outliers, and track trends over time. The insights derived from this process can then be communicated to stakeholders through written reports or presentations.

### 3.2 Our Framework

Following the main job scope of a data analyst, we describe our task setting below. As illustrated in Figure 1, given a business-related question and one or more relevant database tables with their schema, we aim to extract the required data, generate a figure for visualization and provide some analysis and insights.

To tackle the above task setting, we design an end-to-end framework. With GPT-4's abilities in context understanding, code generation, and data storytelling being demonstrated, we aim to use GPT-4 to automate the whole data analytics process, following the steps shown in Figure 1. Basically, there are three steps involved: (1) code generation (shown in blue arrows), (2) code execution (shown in orange arrows), and (3) analysis generation (shown in green arrows). The algorithm of our framework is shown in Algorithm 1.

**Algorithm 1** GPT-4 as a data analyst

---

**Require:** Question $q$; Database schema $s$; Database table $t$; Online $o$
**Require:** Instruction prompts for code generation $p_{code}$, analysis generation $p_{analysis}$
**Require:** LLM $f(\cdot)$; LM decoding temperature $\tau$
**Require:** An external knowledge retrieval model $g(\cdot)$
**Require:** Python compiler $h(\cdot)$
   $Q, C \leftarrow f(q, s, p_{code}, \tau)$         ▷ Generate SQL query (Q) and Python code (C).
   $D, G \leftarrow h(Q, C, s, t)$         ▷ Execute code to get data (D) and graph (G).
   **if** $o$ is *true* **then**         ▷ Only use online information when instructed.
      $I \leftarrow g(q, D)$         ▷ Query information from external knowledge source.
      $A \leftarrow f(q, p_{analysis}, D, I, \tau)$   ▷ Generate analysis (A) from data (D) and online information (I).
      **return** D, G, A
   **else if** $o$ is *false* **then**
      $A \leftarrow f(q, p_{analysis}, D, \tau)$         ▷ Generate analysis (A) from data (D).
      **return** D, G, A
   **end if**

---

Question: [question]

conn = sqlite3.connect([database file name])

[database schema]

Write Python code to select relevant data and draw the chart. Please save the plot to "figure.pdf" and save the label and value shown in the graph to "data.txt".

Table 1: Prompt for the first step in our framework: code generation. Text in blue: the specific question, database file name and database schema.

**Step 1: Code Generation.** The input of the first step contains a question and database schema. The goal here is to generate the code for extracting data and drawing the figure in later steps. We utilize GPT-4 to understand the questions and the relations among multiple database tables from the schema. Note that only the schema of the database tables is provided here due to data security reasons. The massive raw data is still kept safe offline, which will be used in the later step. The designed prompt for this step is shown in Table 1. By following the instructions, we can get a piece of Python code containing SQL queries. An example code snippet generated by GPT-4 is shown in Appendix A.

**Step 2: Code Execution.** As mentioned earlier in the previous step, to maintain data safety, we execute the code generated by GPT-4 offline. The input in this step is the code generated from Step 1 and the raw data from the database, as shown in Figure 1. By locating the data directory using

Question: [question]

[extracted data]

Generate analysis and insights about the data in 5 bullet points.

Table 2: Prompt for the third step in our framework: analysis generation. Text in blue: the specific question and the extracted data as shown in "data.txt".

"conn = sqlite3.connect([database file name])" as shown in Table 1 in the code, the massive raw data is involved in this step. By executing the Python code, we are able to get the chart in "figure.pdf" and the extracted data saved in "data.txt".

**Step 3: Analysis Generation.** After we obtain the extracted data, we aim to generate data analysis and insights. To make sure the data analysis is aligned with the original query, we use both the question and the extracted data as the input. Our designed prompt for GPT-4 of this step is shown in Table 2. Instead of generating a paragraph of description about the extracted data, we instruct GPT-4 to generate the analysis and insights in 5 bullet points to emphasize the key takeaways. Note that we have considered the alternative of using the generated figure as input as well, as the GPT-4 technical report (OpenAI, 2023) mentioned it could take images as input. However, this feature was not open to the public at the time of this paper was written. Since the extracted data essentially contains at least the same amount of information

as the generated figure, we only use the extracted data here as input for now. From our preliminary experiments, GPT-4 is able to understand the trend and the correlation from the data itself without seeing the figures.

In order to make our framework more practical such that it can potentially help human data analysts boost their daily performance, we add an option of utilizing external knowledge sources, as shown in Algorithm 1. Since the actual data analyst role usually requires relevant business background knowledge, we design an external knowledge retrieval model $g(\cdot)$ to query real-time online information (I) from an external knowledge source (e.g. Google). In such an option, GPT-4 takes both the data (D) and online information (I) as input to generate the analysis (A).

## 4 Experiments

### 4.1 Dataset

Since there is no exact matching dataset available, we select the most relevant one, known as the NvBench dataset. We randomly choose 1000 questions from various domains, featuring different chart types and difficulty levels, to conduct our main experiments. The chart types cover bar, stacked bar, line, grouping line, scatter, grouping scatter and pie. The difficulty levels include: easy, medium, hard and extra hard. The domains include sports, artists, transportation, apartment rentals, colleges, etc. On top of the existing NvBench dataset, we additionally use our framework to write insights drawn from data in 5 bullet points for each instance and evaluate the quality using our self-designed evaluation metrics.

### 4.2 Evaluation

To comprehensively investigate the performance, we carefully design several human evaluation metrics to evaluate the generated figures and analysis separately for each test instance.

**Figure Evaluation**   We define 3 evaluation metrics for figures:

- correctness: is the data and information shown in the figure correct?
- chart type: does the chart type match the requirement in the question?
- aesthetics: is the figure aesthetic and clear without any format errors?

| Metrics | | Group 1 | Group 2 | Average |
|---|---|---|---|---|
| Figure | Correctness | 0.77 | 0.78 | 0.78 |
| | Chart Type | 0.99 | 1.00 | 0.99 |
| | Aesthetics | 2.48 | 2.51 | 2.50 |
| Data Analysis | Correctness | 0.94 | 0.94 | 0.94 |
| | Complexity | 2.30 | 2.28 | 2.29 |
| | Alignment | 1.00 | 1.00 | 1.00 |
| | Fluency | 3.00 | 3.00 | 3.00 |

Table 3: Performance of GPT-4 as a data analyst.

The information correctness and chart type correctness are calculated from 0 to 1, while the aesthetics score is on a scale of 0 to 3.

**Analysis Evaluation**   For each bullet point generated in the analysis and insight, we define 4 evaluation metrics as below:

- correctness: does the analysis contain wrong data or information?
- alignment: does the analysis align with the question?
- complexity: how complex and in-depth is the analysis?
- fluency: is the generated analysis fluent, grammatically sound and without unnecessary repetitions?

We grade the correctness and alignment on a scale of 0 to 1, and grade complexity and fluency in a range between 0 to 3.

To conduct human evaluation, 6 professional data annotators are hired from two data annotation companies to annotate each figure and analysis bullet points on the evaluation metrics described above following the detailed annotation guidelines shown in Appendix B. The annotators are fully compensated for their work. Each data point is independently labeled by two different annotators.

### 4.3 Main Results

**GPT-4 performance.**   Table 3 shows the performance of GPT-4 (gpt-4-0314) as a data analyst on 1000 samples. We show the results of each individual evaluator group and the average scores between these two groups. For chart-type correctness evaluation, both evaluator groups give almost full scores. This indicates that for such a simple and clear instruction such as "draw a bar chart", "show a pie chart", etc., GPT-4 can easily understand its meaning and has background knowledge about what the chart type means, so that it can plot

| Annotator | Samples | Figure | | | | Data Analysis | | | | |
|---|---|---|---|---|---|---|---|---|---|---|
| | | Correctness | Chart Type | Aesthetics | Time (s) | Correctness | Complexity | Alignment | Fluency | Time (s) |
| Senior | 30 | 0.79 | 0.96 | 2.96 | 472 | 0.98 | 2.01 | 0.98 | 2.98 | 324 |
| GPT-4 | | 0.73 | 0.96 | 2.41 | 59 | 0.82 | 2.18 | 1.00 | 3.00 | 40 |
| Junior | 30 | 0.66 | 0.96 | 2.66 | 645 | 0.95 | 1.98 | 0.86 | 3.00 | 388 |
| GPT-4 | | 0.71 | 0.98 | 2.75 | 50 | 0.94 | 2.32 | 1.00 | 3.00 | 34 |
| Intern | 40 | 0.74 | 0.91 | 2.40 | 648 | 0.86 | 1.59 | 1.00 | 3.00 | 173 |
| GPT-4 | | 0.73 | 0.97 | 2.45 | 55 | 0.91 | 2.28 | 1.00 | 3.00 | 33 |

Table 4: Overall comparison between several senior/junior/intern data analysts and GPT-4 on 100 random examples in total. Time spent is shown in seconds (s).

the figure in the correct type accordingly. In terms of aesthetics score, it can get 2.5 out of 3 on average, which shows most of the figures generated are clear to the audience without any format errors. However, for the information correctness of the plotted figures, the scores are not so satisfactory. We manually check those figures and find most of them can roughly get the correct figures despite some small errors. As shown in Appendix B, our evaluation criteria are very strict, such that as long as any data or any label on the x-axis or y-axis is wrong, the score has to be deducted. Nevertheless, it has room for further improvement.

For analysis evaluation, both alignment and fluency get full marks on average. It verifies that generating fluent and grammatically correct sentences is definitely not a problem for GPT-4. We notice the average correctness score for analysis is much higher than the correctness score for figures. This is interesting that despite the wrong figure generated, the analysis could be correct. This is because, as mentioned, most of the "wrong" figures only contain some small errors. Thus, only 1 or 2 out of the 5 bullet points related to the error parts from the figures may be generated incorrectly, while most of the bullet points can be generated correctly. In terms of the complexity scores, 2.29 out of 3 on average is reasonable and satisfying. We will show a few cases and discuss more on the complexity scores in Section 4.4.

**Comparison between human data analysts and GPT-4.** To further answer our research question, we hire professional data analysts to do these tasks and compare them with GPT-4 comprehensively. The profiles of the data analysts are described in Appendix C. We fully compensate them for their annotation. Table 4 shows the performance of data analysts of different expert levels from different backgrounds compared to GPT-4. Overall speak-

| Source | Level | Median/Average Annual Salary (USD) | Cost per instance (USD) |
|---|---|---|---|
| levels.fyi | Senior DA | 90,421 | 9.92 |
| | Entry Level DA | 37,661 | 5.36 |
| Glassdoor | Senior DA | 86,300 | 9.47 |
| | Junior DA | 50,000 | 7.12 |
| | Intern DA | 14,400 | 1.63 |
| Our Annotation | Senior DA | - | 11.00 |
| | Junior DA | - | 7.00 |
| | Intern DA | - | 2.00 |
| GPT-4 | | - | **0.05** |

Table 5: Cost comparison from different sources.

ing, GPT-4's performance is comparable to human data analysts, while the superiority varies among different metrics and human data analysts.

Among different levels of human data analysts, overall speaking, the senior group performs the best, followed by the junior group, and finally the interns, especially on the analysis correctness and complexity. Comparing human data analysts with GPT-4, we can notice that GPT-4 outperforms both junior and intern data analysts on most of the metrics, while still having some gap with senior data analysts on three metrics: figure correctness, figure aesthetics and analysis correctness.

Apart from the comparable performance between all data analysts and GPT-4, we can notice the time spent by GPT-4 is much shorter than human data analysts. Table 5 shows the cost comparison from different sources. We obtain the median annual salary of data analysts in Singapore from level.fyi[2] and the average annual salary of data analysts in Singapore from Glassdoor[3]. We assume there are around 21 working days per month and

---
[2] https://www.levels.fyi/
[3] https://www.glassdoor.sg/

| | |
|---|---|
| **Question** | Please list the proportion number of each winning aircraft. |
| **SQL Query** | SELECT
a.Aircraft, COUNT(m.Winning_Aircraft) as wins
FROM aircraft a
JOIN match m
ON a.Aircraft_ID = m.Winning_Aircraft
GROUP BY a.Aircraft
ORDER BY wins DESC |
| **Figure** | 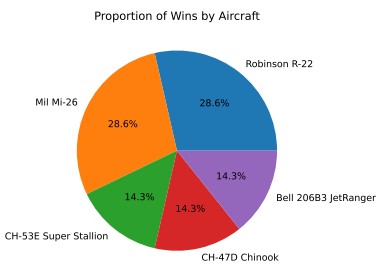 |
| **Analysis** | 1. The Robinson R-22 and Mil Mi-26 are the most successful aircraft in the dataset, with each having 2 wins.
2. The CH-53E Super Stallion, CH-47D Chinook, and Bell 206B3 JetRanger each have 1 win, making them less successful compared to the Robinson R-22 and Mil Mi-26.
3. The Robinson R-22 and Mil Mi-26 together account for 50% of the total wins, indicating their dominance in the competition.
4. The dataset includes a diverse range of aircraft, from helicopters like the CH-53E Super Stallion to smaller aircraft like the Robinson R-22, showcasing the variety of winning aircraft.
5. Despite the variety of aircraft, the data suggests that certain models, such as the Robinson R-22 and Mil Mi-26, have a higher likelihood of winning, potentially due to their design, performance, or other factors. |

Table 6: Case study by GPT-4.

the daily working hour is around 8 hours, and calculate the cost per instance in USD based on the average time spent by data analysts from each level. We pay the data analysts based on the market rate accordingly, to roughly match the median or average salaries from two sources. Specifically, we discuss the pay with each data analyst case by case. For our annotation, the cost of GPT-4 is approximately 2.5% of the cost of an intern data analyst, 0.71% of the cost of a junior data analyst and 0.45% of the cost of a senior data analyst.

## 4.4 Case Study

**Case by GPT-4.** In the case shown in Table 6, GPT-4 is able to generate a Python code containing the correct SQL query to extract the required data,

| | |
|---|---|
| **Question** | List the position of players and the average number of points of players of each position. Visualize by bar chart, and could you sort by the total number in ascending? |
| **Figure** | 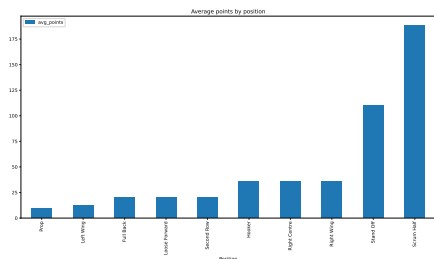 |
| **Analysis** | 1. There are 10 positions. Based on the names, this dataset is about Rugby. Rugby is a group sport that is commonly seen in the US, Canada and Commonwealth countries.
2. Stand Off and Scrum Half are having more than 100 average points while the rest are below 40. This is a bit surprising as usually the Right Wing and Left Wing are the star scorers.
3. Prop has the lowest average points of only 9.33, as Prop's main role is to be in the first row and push against the opposition's Prop.
4. Full Bck, Loose Forward and Second Row has the same points (20), while Hooker, Right Centre and Right Wing has the same points (36).
5. Overall the dataset is slightly different from the general understanding. If there's no data issue, it might worth to reevaluate the players and think about a better position assignment. |

Table 7: Case Study 3 by Senior Data Analyst 2.

and to draw a proper and correct pie chart according to the given question. In terms of the analysis, GPT-4 is capable of understanding the data by conducting proper comparisons (e.g., "most successful", "less successful", "diverse range"). In addition, GPT-4 can provide some insights from the data, such as: "indicating their dominance in the competition". These aforementioned abilities of GPT-4 including context understanding, code generation and data storytelling are also demonstrated in many other cases. Furthermore, in this case, GPT-4 can also make some reasonable guess from the data and its background knowledge, such as: "potentially due to their design, performance, or other factors". However, in another case shown in Appendix D, we notice some numerical errors done by GPT-4, which is very likely due to its issue of hallucination.

**Case by the senior data analyst.** As shown in Table 7, we can notice that this expert human data analyst can understand the requirement, write the code to draw the correct bar chart, and analyze the

extracted data in bullet points. Apart from this, we can summarize three main differences with GPT-4. First, different from GPT-4, the human data analyst can express the analysis with some personal thoughts and emotions. For example, the data analyst mentions "This is a bit surprising ...". In real-life business, personal emotions are important sometimes. With the emotional phrases, the audience can easily understand whether the data is as expected or abnormal. Second, the human data analyst tends to apply some background knowledge. For example, as shown in Table 7, the data analyst mentions "... is commonly seen ...", which is more natural during a data analyst's actual job. While GPT-4 usually only focuses on the extracted data itself, an experienced data analyst is easily linked with one's background knowledge. However, this might be the reason causing the slightly lower alignment scores in Table 4. To mimic a human data analyst better, in our framework, we add an option of using Google search API to extract real-time online information when generating data analysis. We explain our additional experiment integrating the optional online information in Section 4.5. Third, when providing insights or suggestions, a human data analyst tends to be conservative. For instance, in the 5th bullet point, the human data analyst mentions "If there's no data issue" before giving a suggestion. Unlike humans, GPT-4 usually directly provides the suggestion in a confident tone without mentioning its assumptions.

## 4.5 Additional Experiments

**More Practical Questions.** The questions in the experiments above are randomly selected from the NvBench dataset. Although the questions indeed cover a lot of domains, databases, difficulty levels and chart types, they are still somewhat too specific according to human data analysts' feedback. The existing questions usually contain information such as a specific correlation between two variables, and a specific chart type. In a more practical setting, the requirements are more general, which requires a data analyst to formulate a specific question from the general business requirement, and to determine what kind of chart would present the data better.

Therefore, we carefully design five practical and general questions that are acknowledged by a few senior data analysts. To evaluate the comprehensive abilities such as the problem formulation ability of GPT-4, we compare the results among GPT-

4, a senior data analyst and a junior data analyst. The detailed results are shown in Appendix E. For such practical and general questions, the senior data analyst and GPT-4 perform much better than the junior data analyst. The performances of the senior data analyst and GPT-4 are basically on par with each other.

**Online Information Integration.** In Figure 1, we show the optional input of external knowledge in our proposed framework. In some cases, data analysts are not only required to interpret the data in the databases, but also to understand and integrate some industry background knowledge. For such questions, we design an optional module that queries online information from Google and generates the data analysis with the incorporation of the online information. Through our preliminary experiments, this module helps GPT-4 to combine additional knowledge. We show one case in Appendix F.

## 4.6 Findings and Discussions

Generally speaking, GPT-4 can perform comparable to a data analyst from our preliminary experiments, while there are still several issues to be addressed before we can reach a conclusion that GPT-4 is a good data analyst. First, as illustrated in Section 4.4 and Appendix D, GPT-4 still has hallucination problems, which is also mentioned in GPT-4 technical report (OpenAI, 2023). Data analysis jobs not only require those technical skills and analytics skills, but also requires high accuracy to be guaranteed. Therefore, a professional data analyst always tries to avoid those mistakes including calculation mistakes and any type of hallucination problems. Second, before providing insightful suggestions, a professional data analyst is usually confident about all the assumptions. Instead of directly giving any suggestion or making any guess from the data, GPT-4 should be careful about all the assumptions and make the claims rigorous.

## 5 Conclusions

The potential for large language models (LLMs) like GPT-4 to replace human data analysts has sparked a controversial discussion. However, there is no definitive conclusion on this topic yet. This study aims to answer the research question of whether GPT-4 can perform as a good data analyst by conducting several preliminary experiments.

We design a framework to prompt GPT-4 to perform end-to-end data analysis with databases from various domains and compared its performance with several professional human data analysts using carefully-designed task-specific evaluation metrics. Our results and analysis show that GPT-4 can outperform an intern data analyst or a junior data analyst, and can achieve comparable performance to a senior data analyst, but further studies are needed before concluding that GPT-4 can replace data analysts.

## 6 Limitations

**More Experiments.** As mentioned in Section 4.5, the questions from the NvBench dataset contain very specific information, which is somewhat disconnected from real work scenarios. In terms of the broader questions that are more closely related to real work scenarios, only 5 questions are designed and evaluated in this work. Our next step is to collect more practical and general questions to further test the problem formulation ability of GPT-4.

We did not systematically conduct a large number of experiments using online information as well. The reason is similar to the above. The original questions from the NvBench dataset largely depend on the data stored in the database and rarely require additional knowledge. Therefore, we leave the design of such open questions to future work.

**Choice of Annotators.** The quantity of human evaluation and data analyst annotation is relatively small due to budget limitations. For human evaluation, we strictly select those professional evaluators in order to give better ratings. They have to pass our test annotations for several rounds before starting the human evaluation. For the selection of data analysts, we are even more strict. We verify if they really had data analysis working experience, and make sure they master those technical skills before starting the data annotation. However, since hiring a human data analyst (especially for those senior and expert human data analysts) is very expensive, we can only find a few data analysts and ask them to do a few samples.

## Ethics Statement

The purpose of this work is not to replace the data analyst role or to create anxiety. Instead, we would like to explore the potential of GPT-4 to aid human data analysts in more efficient working.

## Acknowledgements

We would like to thank our data annotators and data evaluators for their hard work. Especially, we would like to thank Mingjie Lyu for the fruitful discussion and feedback.

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

# A Example Code

```
1    import sqlite3
2    import matplotlib.pyplot as plt
3
4    # Connect to the database
5    conn = sqlite3.connect("aircraft.sqlite")
6    cursor = conn.cursor()
7
8    # Query to get the count of each winning aircraft
9    query = '''
10   SELECT a.Aircraft, COUNT(m.Winning_Aircraft) as wins
11   FROM aircraft a
12   JOIN match m ON a.Aircraft_ID = m.Winning_Aircraft
13   GROUP BY a.Aircraft
14   ORDER BY wins DESC
15   '''
16
17   # Execute the query and fetch the results
18   cursor.execute(query)
19   results = cursor.fetchall()
20
21   # Extract the labels and values for the chart
22   labels = [row[0] for row in results]
23   values = [row[1] for row in results]
24
25   # Save the labels and values to data.txt
26   with open("data.txt", "w") as f:
27       for label, value in zip(labels, values):
28           f.write(f"{label}: {value}\n")
29
30   # Create the pie chart
31   plt.pie(values, labels=labels, autopct="%1.1f%%")
32   plt.title("Proportion of Wins by Aircraft")
33
34   # Save the chart to figure.pdf
35   plt.savefig("figure.pdf")
36
37   # Close the database connection
38   conn.close()
```

Figure 2: An example of a complete code generated by GPT-4. This code is to answer the question shown in Table 6.

Figure 2 shows an example code generated by GPT-4. First, we can notice that GPT-4 is capable of writing SQL queries with several commands, such as JOIN, GROUP BY, ORDER BY to extract the required data. Second, GPT-4 knows how to use multiple Python packages including sqlite and matplotlib, which help to connect the databases and draw charts respectively. Third, GPT-4 can understand the requirement in the question to save the data and figure it into the correct files accordingly. Last but not least, it can also generate comments understandable by readers, which is aligned with the goal of helping human data analysts boost their daily performance. In the case when the wrong code is generated, a human analyst can easily understand which part goes wrong with the aid of the comments.

# B Detailed Annotation Guidelines

In this section, we present our detailed annotation guidelines for human evaluators.

## B.1 Figure Evaluation

For the figures generated by the model, scores will be given based on the following three criteria, using the correct figure (and correct data) as a reference:

**Information Correctness.** The information correctness can be chosen from 0, 0.5 and 1. First, if the information is correct, 1 point is awarded. Second, if there are minor errors, 0.5 points are awarded. The minor errors mean that the data is mostly correct, but missing one or two data points or showing indexes instead of x-axis/y-axis names, and it does not affect the overall data conclusion. Third, if any important data is incorrect, no points are awarded. The errors include data errors, extra data, missing data, incorrect x-axis/y-axis names, etc. Errors do not include inconsistent color, inaccurate data, inconsistent order, etc.

**Chart Type Correctness.** Since chart type is pretty straightforward, the scores will be binary. If it matches the chart type required in the question, 1 point is awarded; otherwise, 0 point is awarded. For example, if a pie chart is required in the question, but a line chart is drawn, 0 point is awarded.

**Aesthetics.** The score of this metric is on a scale of 0 to 3 points. If all information is clear, it will receive full marks (3 points). If there are minor format issues, 2 points are awarded. If it affects the reader's understanding to some extent, 1 point is awarded. If it seriously affects reading, 0 points are awarded. Subjectivity is relatively high for this metric, and thus we show the annotators a few examples.

## B.2 Data Analysis Evaluation

For each data analysis bullet point generated by the model, we evaluate it from the following four aspects based on the correct data and the correct figure.

**Correctness.** The scores of this metric are binary. The sentence gets 1 point if the information in the sentence is correct. If the sentence contains any false information, it will get a 0 score.

**Alignment.** The scores of this metric are binary as well. The bullet point gets 1 point if it is relevant to the question, and 0 points if irrelevant.

**Complexity.** The score of complexity is o a scale of 0 to 3 points. The bullet point gets 0 points if it is a general description. For example: "This is a bar chart, the x-axis represents ..., the y-axis represents ...". This information is considered very general, which can be obtained without seeing the actual data. The bullet point gets 1 point for directly visible data points or information. For example, "the quantity on wednesday reached 50.", "There are a total of 5 data points.", etc. The bullet point gets 2 points for analysis obtained by comparison or calculation. For example, "the range of certain data is from 23 to 699". This actually includes which one has the highest and lowest scores, and is obtained by comparison, so it is labeled as 2 points. Similarly, another example is, "Wednesday has the highest score, reaching 50". Other types of examples are: "The data increased from 5 to 8", "the sum of A and B on Wednesday is 67", "there is a positive correlation trend". Lastly, The bullet point gets 3 points if the sentence has some insights. For example, "the number on Wednesday reached 50, indicating/suggesting ...". Most of the keywords used are "indicates/suggests/shows", which can basically be worth 3 points. The premise is that the insight should be about the content of the figure. It can be an insight obtained from specific numbers or the positive/negative correlation/trend of the x-axis/y-axis variables. The insights generally can obtain some useful conclusions. ii.

**Fluency.** Fluency and Grammar soundness is on a scale of 0 to 3 points. (a) Very smooth, no grammar errors: 3 points. (b) Some minor grammar or spelling errors that do not affect understanding: 2 points. (c) Some grammar errors that affect understanding to some extent: 1 point. (d) Serious grammar errors that significantly affect reading and understanding: 0 points.

## C   Profiles of Our Data Analysts

We hire 5 data analysts of different levels in total, 2 of them are senior data analysts, 2 of them are junior data analysts and 1 of them is an intern data analyst. One senior data analyst has more than 6 years of data analysis working experience in the finance industry. The other senior data analyst works in the internet industry as a data analyst for over 5 years. One junior data analyst has data analysis working experience in a consulting firm within 2 years. The other junior data analyst works in the supply chain industry as a data analyst for around

2 years. The intern junior data analyst is doing a data analysis internship at a consulting firm.

## D   More Case Studies

| | |
|---|---|
| **Question** | Show me about the correlation between Height and Weight in a scatter chart. |
| **Figure** | 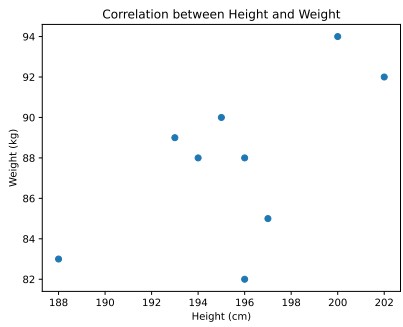 |
| **Analysis** | 1. The data provided consists of 9 data points, each representing the height and weight of an individual. 2. The height range in the dataset is from 188 cm to 202 cm, while the weight range is from 82 kg to 94 kg. 3. The average height of the individuals in the dataset is approximately 195.3 cm, and the average weight is approximately 88.1 kg. 4. There seems to be a positive correlation between height and weight, as taller individuals generally have higher weights. 5. A scatter chart would help visualize this correlation, with height on the x-axis and weight on the y-axis. The data points would likely form an upward trend, indicating that as height increases, weight also tends to increase. |

Table 8: Another Case Study by GPT-4.

Table 8 shows another question addressed by GPT-4. Again, GPT-4 is able to extract the correct data, draw the correct scatter plot and generate reasonable analysis. Although most of the bullet points are generated faithfully, if we read and check carefully, we can notice the numbers of the average height and weight are wrong. Apart from the well-known hallucination issue, we suspect that GPT-4's calculation ability is not strong, especially for those complex calculations. We also notice this issue in several other cases. Although GPT-4 generates the analysis bullet points in a very confident tone, but the calculation is sometimes inaccurate.

## E   More Practical Questions

In this section, we present 5 more practical questions that do not have clear or specific requirements. The questions designed are more likely to be open

| Question | Junior DA | Senior DA | GPT-4 |
|----------|-----------|-----------|-------|
| 1 | 1 | 3 | 2 |
| 2 | 1 | 3 | 2 |
| 3 | 1 | 2 | 3 |
| 4 | 1 | 2 | 3 |
| 5 | 1 | 3 | 2 |

Table 9: Scores of 5 more practical questions.

questions, which do not have one fixed answer. This requires the data analysts and GPT-4 to have a good problem formulation ability. We score them based on their ranking of each question. The one who is ranked the first gets a score 3, the second one gets a score of 2 and the last one gets a score of 1. The results are shown in Table 9. We discuss the results of these 5 questions one by one to evaluate how human data analysts and GPT-4 perform.

Table 10 shows the first more practical question. This question simply asks "Which candidate should we accept?" without specifying the exact requirements for candidate acceptance. We rank the senior data analyst's answer the first among these 3. Instead of only considering the support rate, the senior data analyst also considered the oppose rate provided in the database, and proposed a new metric named net polarity rate. GPT-4 gave the answer by sorting on the support rate. However, GPT-4 mentioned other candidates could be considered if additional factors are taken into account. This indicates the potential of GPT-4 to be trained to be a professional data analyst who has comprehensive thinking.

The results of the second question are shown in Table 11. All 3 annotators gave the same answer by ranking the students based on their average grades. However, the junior data analysts did not specify the names of the students and wrote a few irrelevant analysis bullet points, thus ranked the last by us. In contrast, the human senior data analyst explained the reason for choosing this metric clearly.

Among the results of the third question shown in Table 12, we rank the GPT-4 the first, followed by the senior data analyst and finally the junior data analyst. Both GPT-4 and the senior data analysts analyzed the data based on different regions. Given the limited employee data, GPT-4 still mentioned the potential possibilities of cost reduction in 3 bullet points, while the senior data analyst mentioned it in the last bullet point.

The results of the fourth question are shown in

Table 13. All three annotators provided the same answer. GPT-4 is ranked first because it provided more insights in most of the bullet points. The senior data analyst is ranked second because he/she suggests another metric to measure the popularity of the rooms. Although the junior analyst gave the same answer, 3 out of 5 bullets are irrelevant to the question.

Table 14 shows the results of the last practical question. The senior data analyst's answer is no doubt to be the best, as the figure is clearer and the bullet points are more reasonable than the other two. For the junior data analyst and GPT-4, the answers mentioned in the analysis cannot be seen clearly from the figures. The junior data analyst is ranked last as he/she wrote a few irrelevant analyses again.

## F  Online Information Integration

Table 15 shows one case done by GPT-4 with and without integrating online information. When generating analysis without querying online information, the analysis bullet points are purely from the given database. When incorporating external knowledge, GPT-4 is able to generate relevant additional information from online to make the analysis more powerful.

| **Question** | Which candidate should we accept? |
| --- | --- |

**Junior DA**

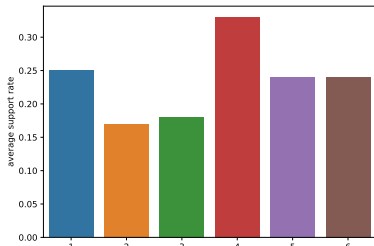

1. Based on data, candidate 4 has the highest average support score.
2. Newsweek Poll has the highest average support score amount three different poll sources.
3. Candidate 2 has extremely high unsure rate given by WNBC/Marist Poll.
4. Candidate 4 has the highest oppose rate.
5. The variance of consider rate is low among all the different rates.

**Senior DA**

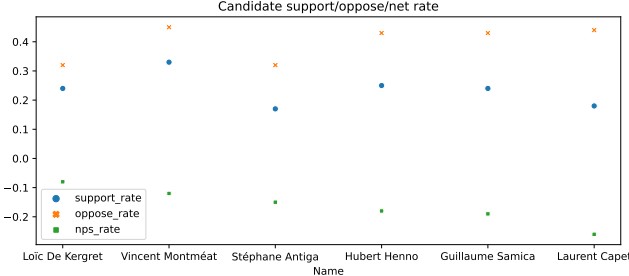

1. Combining the Support Rate, Oppose Rate, and Net Polarity Rate, we should accept Loic De Kergret.
2. Overall, all candidates have a higher oppose rate than the support rate.
3. For the support rate, Loic ranks second. Although Vincent Montmeant has a higher support rate than Loic, Vincent also has a much higher oppose rate.
4. We consider the New polarity rate to be the best metric to take. This metric is calculated as Support Rate substrate the Oppose rate.
5. We think it's important for the candidate to have a high support rate, but also important to have a low oppose rate. Hence the Net Polarity rate is the one we take.

**GPT-4**

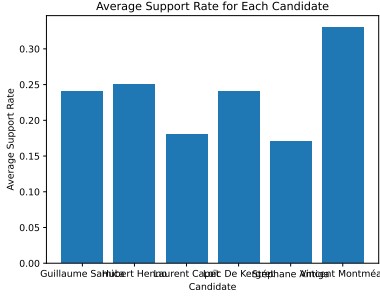

1. Vincent Montméat has the highest average support among the candidates, with a score of 0.33, making him a strong contender for acceptance.
2. Stéphane Antiga and Laurent Capet have the lowest average support, with scores of 0.17 and 0.18, respectively, indicating that they may not be the most suitable candidates for the position.
3. Guillaume Samica and Loïc De Kergret have similar average support scores of 0.24, placing them in the middle of the pack and suggesting that they could be potential alternatives if Vincent Montméat is not selected.
4. Hubert Henno has a slightly higher average support score than Guillaume Samica and Loïc De Kergret, with a score of 0.25, making him another potential alternative candidate.
5. Overall, the data suggests that Vincent Montméat is the most favored candidate, but Hubert Henno, Guillaume Samica, and Loïc De Kergret could also be considered if additional factors need to be taken into account.

Table 10: Comparison of GPT-4, a senior and a junior data analyst on the first more practical questions.

| Question | Choose 3 students who perform well on tests as representatives. |
|---|---|

**Junior DA**

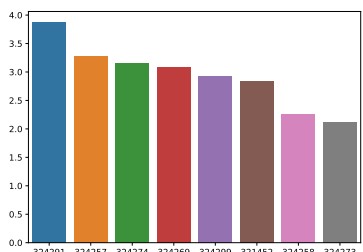

1. Based on GPA, the top 3 students are 324291, 324257, 324274.
2. ACCT department has the most amount of students in dataset, which is 3.
3. Students date of birth are all before 1981.
4. The student hours has a high variance.
5. 3 out of 8 students are transfer students.

**Senior DA**

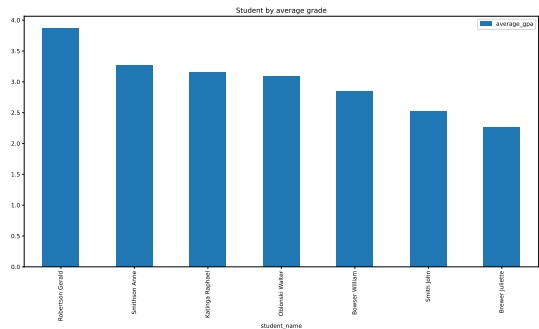

1. We choose to measure the student by their average grade.
2. The calculation method is to take the GPA for every student and take the average. This is to prevent the case where 1 student takes multiple courses.
3. The top 3 students ranked by grade are Robertson Gerald, Smithson Anne, and Katinga Raphael.
4. Robertson Gerald's score is the highest with 3.87, followed by Smithson Anne at 3.27, and Katinga Raphael at 3.15.
5. While Robertson is significantly better than the rest, Smithson and Katinga are not leading the following too much.

**GPT-4**

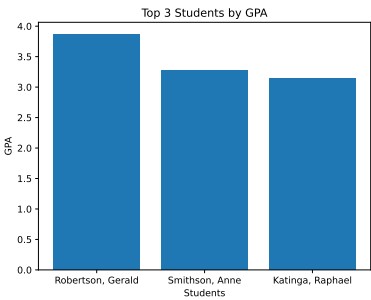

1. The data provided consists of three students' names and their respective test performance scores, presumably on a scale of 0 to 4 or 5.
2. Among the three students, Gerald Robertson has the highest test performance score with a 3.87, indicating that he performs exceptionally well on tests.
3. Anne Smithson has the second-highest score with a 3.27, which suggests that she also performs well on tests, although not as high as Gerald Robertson.
4. Raphael Katinga has the lowest score among the three students with a 3.15, but this score still indicates a relatively good performance on tests.
5. Based on the data, all three students can be considered as representatives of those who perform well on tests, with Gerald Robertson being the top performer, followed by Anne Smithson and Raphael Katinga.

Table 11: Comparison of GPT-4, a senior and a junior data analyst on the second more practical questions.

| **Question** | How to reduce human cost by shifting employees from different departments among these regions? |
|---|---|

**Junior DA**

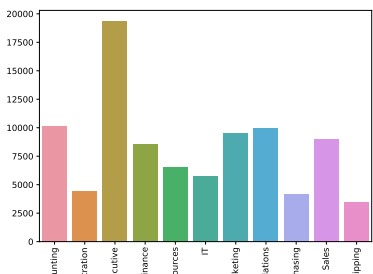

1. Based on the salary of employees across all departments, executive department has the highest salary, followed by accounting and public relations
2. The lowest paid departments are purchasing and administrationand shipping
3. Amoung all the job titles, purchasing clerk, stock clerk and shipper clerk are lowest paid while managers and presidents are highest paid
4. Shipping and sales have the most amount of employees while accounting and administration have lowest amount of employees
5. Based on above, it is shown that it's not feasible to move people from accounting, admin departments to sales, shipping as there are very little people in those department. Instead, finance and IT are higher paid and having more people and could be a good target of restructruing

**Senior DA**

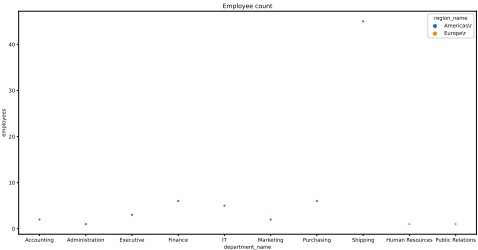

1. We plotted out employee count by the department and colored by region with blue for the American and orange for Europe
2. Europe has only 2 employees, with 1 from Human Resources and 1 from Public Relationship
3. We think it may not be very efficient to set up an EU office with only two employees
4. Usually, Human Resources employees work closely with the rest of the department. Hence we would suggest shifting the Human resource function to the US. For PR, if the marketing focus is EU, it's still ok to keep it in EU
5. However, we would advise the company to consider labour costs in different places as well

**GPT-4**

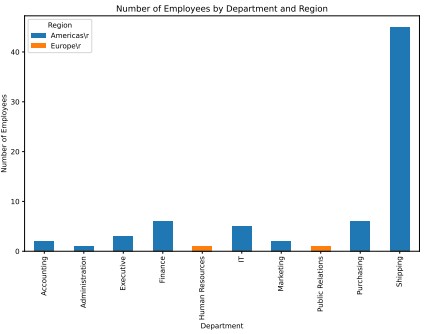

1. The data shows a high concentration of employees in the Shipping department in the Americas region, which could indicate a potential area for cost reduction by redistributing employees to other regions or departments.
2. The Americas region has a diverse range of departments, with Finance and Purchasing having a significant number of employees. This could present an opportunity to shift some employees from these departments to other regions where there is a shortage of staff in similar roles.
3. Europe has a very limited number of departments represented, with only Human Resources and Public Relations having one employee each. This suggests that there may be a need for additional staff in other departments in the Europe region, which could be addressed by transferring employees from the Americas region.
4. The Executive department in the Americas region has three employees, which could be an area to consider for cost reduction by evaluating the necessity of all three positions and potentially consolidating roles or shifting employees to other departments or regions.
5. Overall, the data suggests that there may be opportunities for cost reduction and increased efficiency by evaluating the distribution of employees across departments and regions, and considering strategic shifts of employees to address imbalances and reduce redundancy in certain areas.

Table 12: Comparison of GPT-4, a senior and a junior data analyst on the third more practical questions.

| Question | Which room is more popular? |
|---|---|

**Junior DA**

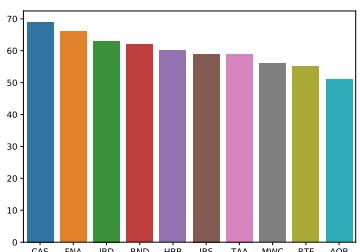

1. 'Convoke and sanguine' and 'Frugal not apropos' and 'immutable before decorum' are the most popular rooms as they have the most amount of reservations.
2. Frugal not apropos has the highest average rate based on reservations.
3. Most of rooms either has traditional or modern decoration.
4. Base price is usually associate with the max number of occupancy.
5. Reservation checkin and checkout data are not in standard format, it's unclear which year it's referring to.

**Senior DA**

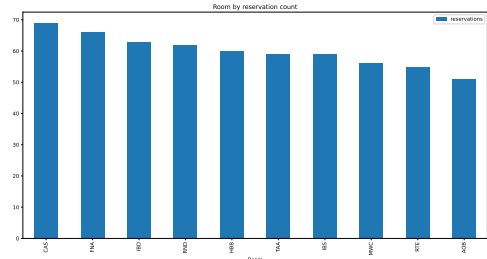

1. CAS is the most popular room if considering reservations, with 69 reservations.
2. It only wins by 3 reservations compared with the room with the second most reservations, FNA.
3. The minimum reserved room is AOB, with 51 records.
4. Overall, the gap between the rooms is not hugely different.
5. Besides reservations, we could also measure the "popularity" by the number of guests.

**GPT-4**

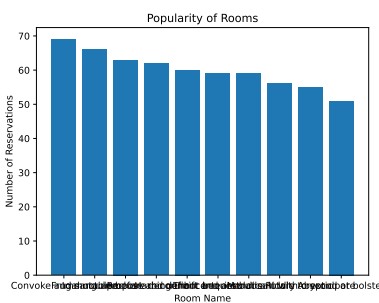

1. The most popular room is "Convoke and sanguine" with 69 visitors, indicating that it may have the most appealing features or activities for guests.
2. The least popular room is "Abscond or bolster" with 51 visitors, suggesting that it may not be as attractive or engaging as the other rooms.
3. The top three rooms ("Convoke and sanguine," "Frugal not apropos," and "Immutable before decorum") have a relatively close number of visitors, ranging from 63 to 69, indicating that they may share similar characteristics that make them popular among guests.
4. The bottom three rooms ("Mendicant with cryptic," "Riddle to exculpate," and "Abscond or bolster") have a more significant difference in visitor numbers, ranging from 51 to 56, suggesting that there may be more variation in their appeal to guests.
5. Overall, there is a relatively even distribution of visitors across the rooms, with only an 18-visitor difference between the most and least popular rooms. This could indicate that guests have diverse preferences and interests, or that the rooms offer a variety of experiences that cater to different tastes.

Table 13: Comparison of GPT-4, a senior and a junior data analyst on the fourth more practical questions.

| Question | A client's budget is 60 and he doesn't like Zinfandel, which wines shall we recommend to him? |
|---|---|

**Junior DA**

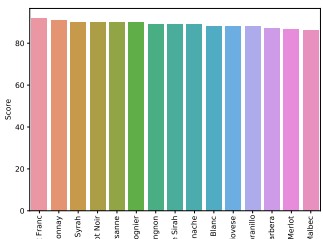

1. We should recommend wine Papas Block to him as this wine is within his budget and not made of Zindandel and has highest Score.
2. The mean score are roughly the same across all different type of grapes.
3. All wines are from California.
4. Most of wines have 'now' as value of drink but there are also some other years appear in this column.
5. Most of the wines are associated with year 2007 which is suspected to be the made year, followed by year 2008 and 2009.

**Senior DA**

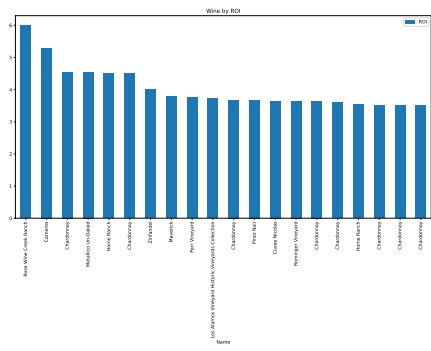

1. The metric we choose to measure ROI. It's calculated as Score/Price. We interpret it as the score value gained for every 1 dollar.
2. Further, we set the minimum score required to be 90.
3. Based on this metric, we would recommend Rose Wine Creek Ranch.
4. Rose Wine Creek Ranch's score is not high (only 90), however, it's only 15 dollars. This is super low compared to the rest.
5. Hence, we would recommend Rose Wine Creek Ranch, as it provides the best value for a dollar.

**GPT-4**

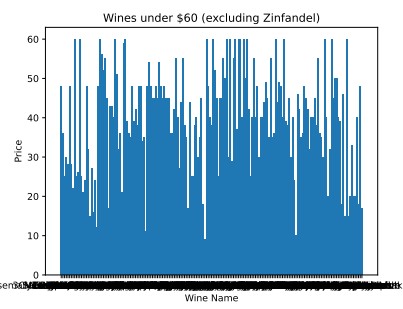

1. The majority of the wines listed are within the client's budget of 60, providing a wide range of options for recommendations.
2. Sauvignon Blanc, Pinot Noir, and Chardonnay are the most frequently listed wine varieties, suggesting that these are popular choices and could be suitable recommendations for the client.
3. The price range for the wines listed is quite broad, from as low as 9 to as high as 60, indicating that there are options available for various preferences and budgets.
4. The client specifically mentioned not liking Zinfandel, and none of the wines listed are Zinfandel, ensuring that all options provided are suitable for the client's taste preferences.
5. Some wines have multiple price points, such as Sauvignon Blanc and Pinot Noir, which could indicate different vintages, vineyards, or quality levels, providing further variety for the client to choose from.

Table 14: Comparison of GPT-4, a senior and a junior data analyst on the fifth more practical questions.

| | |
|---|---|
| **Question** | Combining the data of the phone market in recent years and the database, which phone is more popular? |
| **Figure** | |

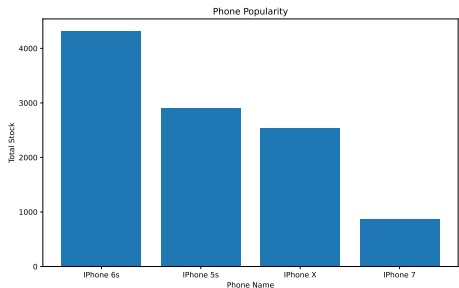

| | |
|---|---|
| Analysis **Without** Online Information | 1. The iPhone 6s is the most popular phone among the given options, with a total stock of 4,324 units.
2. The iPhone 7 has the lowest popularity, with only 874 units in stock, indicating a lower demand compared to other models.
3. The iPhone 5s and iPhone X have moderate popularity, with 2,914 and 2,540 units in stock, respectively.
4. The data suggests that older iPhone models (5s and 6s) still have a significant presence in the market, possibly due to their affordability and continued support from Apple.
5. The lower stock of iPhone 7 could be attributed to its position between the older, more affordable models and the newer, more advanced iPhone X, making it less appealing to consumers. |
| Queried Online Information | 1. This graph shows the market share of mobile operating systems worldwide from May 2022 - May 2023. Apple has 31.12%, Xiaomi has 23.42% and Samsung has ...
2. Android maintained its position as the leading mobile operating system worldwide in the first quarter of 2023 with a market share of 71.4 ...
3. Later, big data—specifically, call detail record data collected from millions of cell phones—was hyped as useful for stopping the disease by tracking ...
4. This research was commissioned to develop a more in depth understanding of smartphone user purchasing behaviour in the UK smartphone market, ...
5. Where does each iPhone rank versus its peers? Here's how we reckon they all compare, from the OG to the latest model.
6. Welcome to the interactive ENISA Smartphone Guidelines Tool. To see information about the domains available click on the names below:. |
| Analysis **With** Online Information | 1. The iPhone 6s is the most popular phone among the given options, with a total stock of 4,324 units.
2. The iPhone 5s comes in second place with a total stock of 2,914 units, indicating that older iPhone models still have a significant presence in the market.
3. The iPhone X, a more recent model, has a lower total stock of 2,540 units, suggesting that it may not be as popular as the older iPhone 6s and 5s models.
4. The iPhone 7 has the lowest total stock of 874 units, indicating that it is the least popular among the given options.
5. Despite the popularity of iPhones in this data, it is important to note that Android maintains a larger market share worldwide (71.4%), which may impact the overall popularity of iPhones in comparison to Android devices. |

Table 15: Comparison of the GPT-4 analysis results with and without online information.