# OpenReview forum: "Is GPT-4 a Good Data Analyst?"
_EMNLP/2023/Conference — EMNLP 2023 Findings_

### Official Review · Reviewer_QeUo · 2023-08-01

**Soundness:** 3

**Excitement:**

3: Ambivalent: It has merits (e.g., it reports state-of-the-art results, the idea is nice), but there are key weaknesses (e.g., it describes incremental work), and it can significantly benefit from another round of revision. However, I won't object to accepting it if my co-reviewers champion it.

**Paper Topic And Main Contributions:**

The authors present an analysis as to how well GPT4 can perform data analysis. They break down the data analysis into three sub tasks of data collection through SQL, data vşusalization and providing insights on the collected data. The experiments are conducted on 1000 queries in natural language from NvBench data set and the results are evaluated by and compared against data analysts.

**Questions For The Authors:**

Q1. How do you eliminate the factor of subjectivity in the proposed evaluation metrics?
Q2. Did you consider different difficulty levels in selecting the queries?

**Reasons To Accept:**

S1. The paper is about a recent topic which has high potential to attrack audience.
S2. A detailed experiment setup  and evaluation is presented.

**Reasons To Reject:**

W1. While the paper presents an interesting analysis, it does not include a contribution towards a new solution.
W2. The proposed evalution metrics are prone to subjective assesment (particularly Aesthetics, Complexity and Fluency). It is not very straightforward how correctness is measured either.
W3. There is no analysis as to extraction of more complex or surprising patterns/insights from the data.

**Reproducibility:**

4: Could mostly reproduce the results, but there may be some variation because of sample variance or minor variations in their interpretation of the protocol or method.

**Reviewer Confidence:**

4: Quite sure. I tried to check the important points carefully. It's unlikely, though conceivable, that I missed something that should affect my ratings.

---

> ### Author Rebuttal · Authors · 2023-08-29
>
> Thank you for the positive feedback and thoughtful review.
>
> **W1:** We designed an end-to-end new framework for the task. Details can be referred to Section 3.2. Our designed framework can outperform junior data analysts and achieve comparable performance to a senior data analyst. On the other hand, the works (e.g. https://aclanthology.org/2023.acl-long.626.pdf accepted at ACL 2023, https://aclanthology.org/2021.acl-long.172.pdf accepted at ACL 2021) presenting interesting analysis without any new solution should not be regarded as flawed. In contrast, they offer valuable insights and analysis from an interesting perspective. These findings contribute to the readers' understanding of a particular model or domain and can also make meaningful contributions to the NLP community.
>
> **W2 & Q1:** In an effort to mitigate the subjectivity of the evaluation process, we have taken a deliberate approach by meticulously designing specific standards for each score of every metric, as outlined in Appendix B. We acknowledge that defining these metrics can be challenging, which is why we have opted for a more limited scale. Using a common scale of 0-5 could introduce greater subjectivity. Metrics such as fluency are commonly employed in tasks like summarization and generation, and their inclusion is essential for evaluating the quality of generated outputs.
>
> **W3:** Thanks for your suggestion on more complex and surprising patterns/insights. Based on our experiments, we have observed that GPT-4 can sometimes analyze correlations, distributions, and outliers within the given data. We will provide illustrative examples in the final version. However, it is important to note that many of the questions in the nvBench dataset are relatively straightforward, where complex insights may not be necessary. Additionally, nvBench is currently the only available dataset that aligns with the steps involved in the data analysis process. In this work, our primary goal is to validate GPT-4's capability to automate the data analysis process, rather than solely focusing on generating surprising insights.
>
> **Q2:** The nvBench paper classified each question into different difficulty levels. Among our 1000 samples, 199 are easy, 392 are medium, 286 are hard, and 123 are extra hard. The scores are shown below:
> |             |             | easy + medium | hard + extra hard |
> |-------------|-------------|:--------------:|:------------------:|
> | Figure      | Correctness |     0.79      |       0.75       |
> |             | Chart Type  |     0.99      |       1.00       |
> |             | Aesthetics  |     2.52      |       2.46       |
> | Data analysis | Correctness |     0.95      |       0.93       |
> |             | Complexity  |     2.32      |       2.25       |
> |             | Alignment   |       1.00       |       1.00       |
> |             | Fluency     |       3.00       |       3.00       |

---

### Official Review · Reviewer_jqtG · 2023-08-04

**Soundness:** 4

**Excitement:**

4: Strong: This paper deepens the understanding of some phenomenon or lowers the barriers to an existing research direction.

**Paper Topic And Main Contributions:**

The paper investigates the ability of GPT-4 to be a data analyst. The model is provided with a question and a database and is tasked to write a python code to obtain the relevant information, execute the code and finally analyze the obtained results to generate insights along with visualization. The performance of GPT-4 is then compared with data analysts at intern, junior and senior levels with GPT-4 outperforming the intern and junior data analyst across several use cases.

**Questions For The Authors:**

How was the correctness determined for practical questions?

GPT-4 was also allowed to search google to gain domain specific knowledge. What kind of search queries were actually made? Are they similar to what a human would make?

Did you find the results to be sensitive to prompt?

**Reasons To Accept:**

Deals with an interesting problem. Nicely written and easy to follow.

Exhaustive experiments were performed to justify the findings.



**Reasons To Reject:**

The paper does not report what happens when the question is open-ended. For example, a data analyst is often given a data without a specific question. The task is to identify patterns/correlations in the dataset. Will GPT-4 be able to deal with that? This is not a reason to reject per se.

The dataset could be made available.

**Reproducibility:**

4: Could mostly reproduce the results, but there may be some variation because of sample variance or minor variations in their interpretation of the protocol or method.

**Reviewer Confidence:**

4: Quite sure. I tried to check the important points carefully. It's unlikely, though conceivable, that I missed something that should affect my ratings.

---

> ### Author Rebuttal · Authors · 2023-08-29
>
> Thanks for the positive feedback and insightful questions.
>
> Regarding your suggestion on **the open-ended question**, this is a meaningful direction. As stated in Section 4.5 and Appendix E, we worked toward this to evaluate whether GPT-4 has good problem formulation ability. Tables 10-14 show the performance. The practical questions here are actually open-ended and do not have definite answers.
>
> **Q1:** We do not have correctness scores for practical questions since they do not have definite answers. In Appendix E Lines 911-918, we mentioned we evaluated the performance based on comparisons among a junior DA, a senior DA, and GPT-4.
>
> **Q2:** In order to guarantee that the retrieved information is highly relevant to the questions, we input both the original question and the processed data as search queries. This strategy resembles human behavior, as humans also tend to use data alongside their questions to achieve more accurate results when performing searches.
>
> **Q3:** The results may differ given different prompts, but overall are not that sensitive to prompts. As highlighted in CoT (Wei et al., 2022) Section 3.4, prompt wording (different terminologies) has little impact on results, as long as the structure remains similar. Furthermore, due to cost concerns, we do not conduct quantitative evaluations on different prompting methods.
>
> We will **make the dataset available**, as stated in Lines 28-29.

---

### Official Review · Reviewer_pggJ · 2023-08-07

**Soundness:** 3

**Excitement:**

3: Ambivalent: It has merits (e.g., it reports state-of-the-art results, the idea is nice), but there are key weaknesses (e.g., it describes incremental work), and it can significantly benefit from another round of revision. However, I won't object to accepting it if my co-reviewers champion it.

**Paper Topic And Main Contributions:**

The paper evaluates the performance of GPT-4 for data analysis using  questions randomly selected from the NvBench benchmark and compares it with data analysts at different experience levels, all with the help of crowd annotators. Results show GPT-4 can perform better than or on par with data analysts.


**Questions For The Authors:**

- Subjective feedback in human-subject experiments is typically elicited on a [Likert scale](https://en.wikipedia.org/wiki/Likert_scale) of 5.  What is the motivation behind using a 4-level (0-3) response scale?

**Reasons To Accept:**

- Addresses an intriguing and high-impact question and offers nuanced discussion.

- Provides substantial initial analysis and obtains results on which future work can be built.

- Acknowledges the importance of open-ended (exploratory) analysis.


**Reasons To Reject:**

- GPT-4  is trained on public Web data and more. So, it is likely already trained on the NvBench benchmark dataset. It needs to be clarified how the data contamination is handled here.

- An essential suggestion by the work is to use external knowledge to mimic a data analyst’s domain understanding and experience. However, it is unclear how Google search provides that beyond the recency of information. GPT-4 is already supposed to encapsulate public web knowledge. Also, details on how Google search is used and how search results are filtered to feed them back to GPT-4 are missing.

- Tables 3 and 4 show integer-valued alignment and fluency scores. Given the large number of crowd ratings, they look infeasible. Showing the standard deviation values and analyzing the statistical significance of differences (for Table 4) would be helpful.

- In practice, data analysts spend most of their time getting data ready for analysis (data preparation). Then, they perform exploratory and confirmatory analyses, determining what questions to ask and finding answers, if possible (data rarely tells clear answers). This process is noisy and iterative, more like a multi-turn conversation than answering one independent question at a time. In many cases, data analysts don’t know which questions to ask.


**Reproducibility:**

4: Could mostly reproduce the results, but there may be some variation because of sample variance or minor variations in their interpretation of the protocol or method.

**Reviewer Confidence:**

4: Quite sure. I tried to check the important points carefully. It's unlikely, though conceivable, that I missed something that should affect my ratings.

---

> ### Author Rebuttal · Authors · 2023-08-29
>
> Thank you for the positive feedback and thoughtful comments.
>
> **W1: data contamination**. First, according to the GPT-4 Technical Report (Page 10), “GPT-4 generally lacks knowledge of events that have occurred after the vast majority of its pre-training data cuts off in September 2021”; while the arXiv version of the nvBench paper was submitted on 24 Dec 2021. Consequently, it is unlikely that data contamination issues would arise due to the chronological order of information. Additionally, in Section 4.5, we created practical questions that were intentionally excluded from GPT-4's exposure. GPT-4 still demonstrates proficiency in answering those questions. Lastly, it is important to note that data contamination remains an open question and a separate research area. In our work, we did not prioritize investigating this aspect.
>
> **W2: using external knowledge**. GPT-4 is designed to encompass public web knowledge, but it does not have access to real-time information, especially after September 2021. Furthermore, its primary focus is often centered on the given data itself, rather than explicitly integrating online information. In the revised version, we will provide more details on how external knowledge is incorporated to address these concerns.
>
> **W3: integer-valued alignment and fluency scores**. Indeed, GPT-4 performs well on these two metrics, achieving full marks. We will add the std and p-values in the revised version.
>
> **W4: practical cases for data analysts**. As stated in Section 4.5, we worked toward this to evaluate whether GPT-4 has good problem formulation ability. However, it is very challenging to comprehensively conduct such an evaluation as explained in the limitation section. In this paper, our primary focus lies on the initial step, specifically when dealing with specific questions. The exploration of more practical scenarios falls within the scope of our ongoing work, which we consider as future research directions.
>
> **Q1: score scales**. In an effort to mitigate the subjectivity of the evaluation process, we have taken a deliberate approach by meticulously designing specific standards for each score of every metric, as outlined in Appendix B. Instead of adhering to the conventional Likert scale with five levels, our intention is to ensure clarity and precision. Through our careful analysis, we determined that utilizing a 4-level response scale adequately captures the necessary nuances for certain metrics, making it sufficient for our purposes.

---

### Official Review · Reviewer_AeQb · 2023-08-12

**Typos Grammar Style And Presentation Improvements:** None at the moment
**Soundness:** 3

**Excitement:**

3: Ambivalent: It has merits (e.g., it reports state-of-the-art results, the idea is nice), but there are key weaknesses (e.g., it describes incremental work), and it can significantly benefit from another round of revision. However, I won't object to accepting it if my co-reviewers champion it.

**Justification For Ethical Concerns:**

None at the moment

**Missing References:**

None at the moment

**Paper Topic And Main Contributions:**

This paper studies an interesting topic about whether GPT can be a good data analyst.  In detail, the authors regard GPT4 as a data analyst to perform end-to-end data analysis with databases from a wide range of domains and propose a framework to do data analyze with GPT. Evaluations by human metrics show the effectiveness of the frametwork and demonstrate GPT-4's ability.

**Questions For The Authors:**

1. Can you conduct more experiments with more human evaluators.

2. Can you provide some result of a wider range of large language models.

**Reasons To Accept:**

1. A sound new research direction to show the ability of GPT-4 to conduct data analyze, which may have further impact to reduce the working load for data analytics.

2. Extensive evaluations with 1000 questions and many case studies.

3. Good written and easy to follow.

**Reasons To Reject:**

1. Limited evaluation with only 6 human evaluators.

2. Only the result for GPT-4 is available, and there is no result for other language models.

**Reproducibility:**

3: Could reproduce the results with some difficulty. The settings of parameters are underspecified or subjectively determined; the training/evaluation data are not widely available.

**Reviewer Confidence:**

3: Pretty sure, but there's a chance I missed something. Although I have a good feel for this area in general, I did not carefully check the paper's details, e.g., the math, experimental design, or novelty.

---

> ### Author Rebuttal · Authors · 2023-08-29
>
> Thank you for the positive feedback and comments.
>
> **W1 and Q1:** As mentioned in the limitation section, we can only do the evaluation on 1000 samples due to the budget limit. Since the evaluation requires some data analysis skills, the evaluator selection was rigorous where they had to pass several rounds of test annotations. We will try to conduct more experiments with more evaluators in future work.
>
> **W2 and Q2:** As the paper title infers, we aim to investigate whether GPT-4 is a good data analyst. GPT-4, being acknowledged as the most advanced model currently available in the market, prompts us to examine the extent of its proficiency. During our preliminary experiments, we experimented with several other commonly used LLMs, such as Claude, but their performance could have been more satisfactory. We will leave the comparison with other LLMs for future work.

---

### Meta-Review · Area_Chair_T4A1 · 2023-09-18

**Recommendation:** 5

**Metareview:**

This paper studies an interesting topic about whether GPT can be a good data analyst. The paper is well-motivated and easy to follow, the experiments are solid, and it has the potential real impact to significantly reduce the workload for data analytics. However, reviewers also point out the paper could add more in-depth analysis, the data and label generated are subjective, the evaluation results have some issues.

---

### Decision · Program_Chairs · 2023-10-07

**Decision:**

Accept-Findings

**Comment:**

This paper studies an interesting topic about whether GPT can be a good data analyst. The paper is well-motivated and easy to follow, the experiments are solid, and it has the potential real impact to significantly reduce the workload for data analytics. However, reviewers also point out the paper could add more in-depth analysis, the data and label generated are subjective, the evaluation results have some issues.